# Machine Learning to Predict Junction Temperature Based on Optical Characteristics in Solid-State Lighting Devices: A Test on WLEDs

**DOI:** 10.3390/mi13081245

**Published:** 2022-08-02

**Authors:** Mohammad Azarifar, Kerem Ocaksonmez, Ceren Cengiz, Reyhan Aydoğan, Mehmet Arik

**Affiliations:** 1EVATEG Center, Ozyegin University, Istanbul 34794, Turkey; mohammad.azarifar@ozu.edu.tr (M.A.); kerem.ocaksonmez@ozu.edu.tr (K.O.); ceren.cengiz@ozu.edu.tr (C.C.); 2Department of Computer Science, Ozyegin University, Istanbul 34794, Turkey; reyhan.aydogan@ozyegin.edu.tr; 3Mechanical Engineering Department of Auburn University, Auburn, AL 36849, USA

**Keywords:** junction temperature, temperature prediction, light emitting diodes, machine learning, solid-state lighting, gradient boosted trees, random forest

## Abstract

While junction temperature control is an indispensable part of having reliable solid-state lighting, there is no direct method to measure its quantity. Among various methods, temperature-sensitive optical parameter-based junction temperature measurement techniques have been used in practice. Researchers calibrate different spectral power distribution behaviors to a specific temperature and then use that to predict the junction temperature. White light in white LEDs is composed of blue chip emission and down-converted emission from photoluminescent particles, each with its own behavior at different temperatures. These two emissions can be combined in an unlimited number of ways to produce diverse white colors at different brightness levels. The shape of the spectral power distribution can, in essence, be compressed into a correlated color temperature (CCT). The intensity level of the spectral power distribution can be inferred from the luminous flux as it is the special weighted integration of the spectral power distribution. This paper demonstrates that knowing the color characteristics and power level provide enough information for possible regressor trainings to predict any white LED junction temperature. A database from manufacturer datasheets is utilized to develop four machine learning-based models, viz., k-Nearest Neighbor (KNN), Radius Near Neighbors (RNN), Random Forest (RF), and Extreme Gradient Booster (XGB). The models were used to predict the junction temperatures from a set of dynamic opto-thermal measurements. This study shows that machine learning algorithms can be employed as reliable novel prediction tools for junction temperature estimation, particularly where measuring equipment limitations exist, as in wafer-level probing or phosphor-coated chips.

## 1. Introduction

The invention of the first nitride-based blue LED by Isamu Akasaki and others, in 1986, revolutionized the general lighting industry [1]. Followed by increased efficiency, manufacturing cost reduction, and the attainment of high color-rendering indexes with the phosphor-based white light generation, white LEDs quickly dominated the general lighting markets [2]. WLEDs soon reached the efficacy of over 100 lm/W inside packages that can handle more than 10 W of input power [3]. Thermal management is still a critical factor for high-power LEDs. The junction temperature (T_j_) rise generated by large self-heating fluxes has shown significant impacts on efficiency, optoelectrical characteristics, and the reliability of LEDs [4,5]. Fundamentally, heat is being generated in the active region of an LED due to non-radiative recombinations, radiation absorption, and Joule heating [6], and in a white LED, in the color conversion composite [7]. Heat from the active region, color conversion composite, and other heat generation zones (e.g., light absorption) should conduct through a complex three-dimensional thermal resistance network, starting from each epitaxial layer to the integrated heat management system and ending at convection to the ambient medium. Thus, in a steady-state condition, the average temperature of the junction depends on thermal conductivity, chip-to-ambient thermal resistance, and local temperatures of nearby heat generation zones (e.g., phosphor composite) [8,9,10,11].

### 1.1. A. Junction Temperature Effect on Emission Characteristic

In an interrelated manner, the T_j_ alters the temperature-dependent radiative and non-radiative recombination of a WLED, as well as carrier leakage from active regions in some instances [12], and ultimately reduces the efficiency of the light generation, known as thermal droop [13]. Emission characteristics of the chip also change by temperature variations due to the temperature dependence of the energy bandgap [14]. This leads to the use of temperature-sensitive optical parameters (TSOPs) for the T_j_ measurement, which are selected parameters from the spectral power distribution (SPD), such as the peak wavelength or spectral bandwidth [15]. The TSOP T_j_ measurement methods have shown their practicality with uninterrupted and non-destructive procedures [16,17,18].

Chhajed et al. [19] calibrated the peak wavelength for UV, blue, green, and red GaN-based LEDs from a forward current of 10 to 100 mA at a temperature of 22 to 120 °C. The temperature coefficients of the dominant wavelength for the blue, green, and red LEDs were determined as 0.0389, 0.0308, and 0.1562 nm/K, respectively. A strong red shift of red AlGaInP LED was also seen in the other study of authors for a trichromatic white LED system [16]. Chen et al. conducted T_j_ measurement experiments and obtained peak wavelength shifts for three different AlGaInP LED arrays [20]. Shifting characteristics investigated at longer, central, and shorter wavelengths have shown that the center wavelength is the most suitable method for determining the junction temperature of an LED array. Tamura et al. [21] analyzed the wavelength shift of an InGaN-based white LED at various temperatures from 20 to 160 °C, and their experiments have shown that blue-light emission from the active layer and yellow-light emission from YAG phosphor formed two different electroluminescence bands. Each band displayed distinct behaviors with changing temperatures. However, similar to previous cases, the T_j_ was calibrated with the blue emission of the chip. Chen et al. [22] have shown a simplified peak wavelength shift variation in a different T_j_ for white LED under different drive currents. Gu et al. [23] selected the point of interest as the lowest energy in the SPD between the peaks of blue and yellow emissions. The ratio of the total radiant energy of white LED to the radiant energy within the blue emission in different junction temperatures has displayed a linear relationship. This transfer function can be used for T_j_ prediction.

Furthermore, the TSOP measurement methods have proven to be a practical approach for measuring the phosphor temperature in an operating white LED. Based on the total emission division of a white LED to a sum of the spectrum of the blue chip and two spectra from the phosphor with a short and a long wavelength band, Yang et al. [24] examined the fitting peak wavelengths and FWHMs of the short and long wavelength bands at the different phosphor temperatures. They have stated that the phosphor temperature can be precisely measured by checking the variations of its related emission spectrum. Like other LEDs, a red shift at higher temperatures was observed.

### 1.2. B. Commonality of SPD Response to Junction Temperature and Input Current Density Change

Fundamentally, it can be seen that one of the outcomes of the T_j_ increase is the red shift and the broadening of the SPD [25], as illustrated in Figure 1a. A red shift happens due to the bandgap reduction at elevated temperatures, studied comprehensively by Wang et al. [26,27] for GaN-based blue LEDs. Recently, similar results were reported at high temperatures and high-brightness GaN on sapphire blue LEDs [28]. It should be noted that the SPD shows a blue shift and broadening with an increasing input current. Li et al. [29] studied the effect of the input current and temperature on the spectral behavior of green InGaN/GaN multi-quantum-well LED and showed that the excitation source can alter the carrier dynamics in the active region. A large blue shift was observed in high-input powers, mainly due to the carrier screening effect due to a weakened piezoelectric field that results in the quantum-confined Stark effect [30]. It is noted that, in general, lighting applications blue shift does not occur as commercial chip packages are tuned to operate at a constant current density. Although LEDs can operate at higher efficiencies in lower current densities (e.g., 10 A/cm^2^), due to inadequate luminous flux per wafer area in these current densities, 35 A/cm^2^ is the widely used input current density value with acceptable internal quantum efficiency [3,31,32]. Overall, it can be stated that TSOPs are unique for each electrical working condition, and because 35 A/cm^2^ is the commonly accepted nominal current density, LED manufacturers provide the TSOP behavior of chips mainly in this current density.

### 1.3. C. Luminous Flux and Correlated Color Temperature, Data Interpretations from SPD

As shown in Figure 1a,b, as the true “fingerprint” of any light source, the SPD can be rendered to how our eyes can see its color via the CIE 1931 color space diagram [33] and how bright it is to our eyes based on the weighted integration in the visible luminous efficiency function [34]. It should also be noted that light intensity also changes depending on the internal quantum efficiency behavior of the LED at each specific temperature and input current [35]. The luminescence of LEDs decreases with a temperature rise, while Meneghini et al. [13] recently presented a tutorial article on the physical origin of this phenomenon. The shape and intensity of the spectral power distribution, in essence, can be interpreted to the correlated color temperature (CCT) and luminous flux (LF).

### 1.4. D. Temperature Dependent Data Presentation by Manufacturers

Schematically outlined in Figure 1c, WLED manufacturers, after the SPD analysis, provide three temperature-dependent optical parameters for their chips: (1) the temperature dependency of the LF, (2) the temperature dependency of the CIE 1931 x coordinate (ccx), and (3) the temperature dependency of the CIE 1931 y coordinate (ccy). As discussed earlier, this information is provided in the current density of 35 A/cm^2^. Presented in Figure 1a–c, these three graphs carry necessary information about the SPD (through chromaticity functions) at each T_j_. Our data analysis of two major chip manufacturers has revealed that the LED upstream industry [36] has reached a concurrence point, where similarities in the responses of the LF, ccx, and ccy to the T_j_ can be seen. It seems that the technology of the developing epitaxy process includes the production of the emitting layer, cladding layer, and buffer layer, and the reflector to the micro-manufacturing process of the design of the electrodes for current spreading, optical microstructures for light extraction, and segmentation are all similar between manufacturers. Overall, the temperature-sensitive optical parameters of the blue chips and phosphors are presenting consistent behavior, while divergence can only be seen from the mid- and downstream LED manufacturing industry. This divergence only changes the thermal resistance network of the LED device, not the temperature sensitivity of the SPD. This seemingly universal behavior can pave the way for a mutual ML algorithm for all WLEDs to predict their T_j_ inside a package or a system, only by knowing their CCT and LF.

### 1.5. E. Predictable Response to Temperature and Possibility of Algorithm Training

In recent years, machine learning (ML) methods have been utilized to predict the SPD and reliability and lifetime prediction of LEDs [37,38]. Lu et al. [39] proposed a lifetime prediction method for WLEDs based on a multidimensional back-propagation (BP) artificial neural network (ANN). For the model, the temperature, electric current, initial chrominance, and initial luminous flux are selected as input neural layers, and the service life of the LED was the expected outcome. Overall, the BP-ANN improved with the Adaboost algorithm and was found to be a promising solution to lower the lifetime prediction error, but the estimation time took longer. Liu et al. [40] employed the BP-ANN to study the multi-physics interrelation between the electrical power, light output, and thermal dissipation for LED systems. For the simplified photo electron–thermal (PET) multi-physics prediction model, the drive current and the temperature were selected as the input layers, while luminous flux, optical power, and electrical power were defined as the output layers, and the model accuracy was updated according to the seven hidden-layer neurons. The BP-ANN model was trained with experimental optical and electrical data at a 0-450 mA drive current and a good agreement between the measured and predicted luminous flux and the optical and electrical power was observed. It was concluded that the PET interrelations can be efficiently made with at least a 6.7 times reduction in the computing resource with the developed ANN model. In their later article [41], a reduced simulation time and higher accuracy in lifetime prediction were reported. The correlation coefficient of 0.99715 between the predicted and training data claimed to be an indicator for the successful prediction of the lifetime and the reliability of a multi-chip LED. 

Furthermore, Chen et al. [42] conducted anomaly detection on the color failure of LEDs using a k-Nearest Neighbor (k-NN) kernel-density-based cluster algorithm. A principal component analysis was carried out for the dimensionality reduction of 24 extracted SPD features. The principal components were divided into clusters using k-NN kernel-density clustering, and anomalies are detected when the test point distance from the centroid of the cluster is larger than the predefined threshold distance. Fan et al. [43] proposed a BP and Genetic Algorithm (GA)-BP ANN for the dynamic SPD prediction of the full spectrum of the white LED operating under different conditions. The method applicability was tested for data outside the experimental dataset and when the training data are small. As a result, a strong dependence between the prediction accuracy and the nature of LED, and the amount of test data was found. With the proposed BP-ANN and GA BP-ANN, the averaged SPD root mean squared error was found to be 6E-5 while the chromaticity difference was 0.002. 

Although ML algorithms are becoming more popular for LED lifetime and SPD prediction, little attention has been paid to LED T_j_ estimation using ML approaches. Merenda et al. [44] introduced an ML algorithm for the T_j_ prediction of an LED. In their study, the T_j_ was evaluated according to the LED part number, aging of the device, and measured current and voltage values. The training dataset was formed by measuring the T_j_ of five LEDs with FVM at different current levels and aging conditions. The regression model was derived from standardized input values with two hidden layers, and L1-L2 regularizations were used to prevent overfitting. Validation studies have shown that using 10 inputs with 84 MHz frequency average interface time, the model can predict the LED T_j_ in 2 ms. Although the proposed algorithm is claimed to predict the Tj with a ±2 °C accuracy in the temperature range of 50-110 °C, the number of samples used for the dataset is not high enough to form a broadly applicable prediction model. 

Considering the LF, ccx, and ccy while building the prediction model for the T_j_ is advantageous as the predictions will be independent of the device thermal resistance. Having information on the emitted light would be enough to perform agile testing of the device in any environment. This article explores the reliability of such an ML method for the T_j_ measurement of commercial WLEDs at their nominal current densities. k-Nearest Neighbor (KNN), Radius Near Neighbors (RNN), Random Forest (RF), and Extreme Gradient Booster (XGB) have been employed for the regression analysis to predict the T_j_. The rest of this paper is organized as follows: Section 2 explains the proposed method for predicting the T_j_ of an LED, and Section 3 reports our experimental findings compared to the predicted results. Finally, Section 4 concludes the paper with the future direction of research. 

## 2. Method

Figure 2 depicts the interconnected processes of data collection, regression, and experiments. For this work, the data provided by LED manufacturers and the data collected through experiments on LEDs produced by different LED manufacturers were used to train and test prediction models. WLEDs from manufacturers other than those from which the data were gathered were chosen for experimental study. T_j_ measurements were based on the forward voltage method with detailed explanation in [45]. First, devices were calibrated using the EVAtherm junction temperature measurement system to determine their k-factor [46]. During the data acquisition test phase, a time-dependent self-heating approach [47] was used to obtain the highest possible number of data points in T_j_, ranging from room temperature until the device reached a steady state. Set-up for the transient test phase can be seen in Figure 2. LEDs were connected to a Keithley 2602B high-precision source meter controlled by data acquisition software and were positioned at the center of a LabSphere Illumia Plus 610 integrating sphere with a 2 m diameter sphere which was painted with a highly reflective coating (reflectivity of 0.98). As shown in Figure 3, as the LED starts to operate with a forward current of I_f_, T_j_ experiences an unsteady temperature rise from room temperature to a steady condition. During this process, a series of source delay measurement (SDM) cycles, with a pulse current of I_f,p_ = 1 mA with a delay time of t_p_ = 3 ms (including 1 ms delay and two 4-wire voltage measurements, each longing for 1 ms) were considered to evaluate time-dependent T_j_. Pulses are delayed with a control interval of t_c_ = 200 ms. Simultaneously, time-dependent optical measurements (ccx, ccy, and LF) with a CDS-610 model detector inside the sphere were recorded. The optical and thermal data were then time-interpolated.

The data obtained from manufactures contain LF, ccx, and ccy corresponding to a specific junction temperature of different LED packages. It is not surprising that, of all the features we could consider from the data sheet, optical parameters appeared to be more important for temperature predictions than electrical and physical properties. In initial tests to note a universal behavior between different manufacturers, correlations between optical features and target (junction temperature) presented significant predictability and thus were chosen for training. It was ensured to obtain broad coverage of cold-to-warm white colors at the different brightness levels in the dataset. If there was insufficient number of instances for a brightness level or color, data collection was continued for such instances. In the end, a balanced dataset covering different brightness and colors was achieved. In the training session, data from over 500 commercial white LED packages (Cree LED and Nichia Corporation) were obtained. Next, data were preprocessed by carrying out digitization, relying on user-supervised automatically generated plot-digitizing algorithms at 1°C interval in the T_j_ range of 25 to 100 °C. Thus, 75 different data instances (i.e., ccx, ccy, LF, and T_j_) per each LED package were recorded. As a result, data instances exceeded the total of 37500. For regression, our target was to predict the temperature of an LED package given its LF, ccx, and ccy.

ML algorithms have proven to successfully handle numerous features, rank their importance, and have high nonlinearity complexity with multivariate relationships [48]. In this study, we aim to apply different regression algorithms and compare their performances to choose the best one for our problem. Accordingly, we trained our dataset with K-Nearest Neighbor (KNN), Radius Near Neighbors (RNN), Random Forest (RF), and Extreme Gradient Booster (XGB) separately. A brief explanation of these algorithms is as follows:

KNN and RRN [49] are seen as “black-box” regressors where functions are predicted from the dataset and are non-parametric, simple yet effective supervised learning algorithms. They are widely used for classification tasks but can also be used for regression. In KNN regression, the prediction is the average of the property values of k-Nearest Neighbors with their weights depending on the given distance functions. The RNN is a simple extension of the KNN. The performance of the regressor highly depends on the number of the nearest neighbors; thus, the accuracy of the regressor can be tuned by the number selected for the close neighbors and the type of distance metric range among the outputs [50]. 

The RF regressor [51] is a supervised learning algorithm that combines the predictions of a collection of decision tree regressors in order to make a more accurate prediction. RF constructs numerous decision trees in parallel and outputs the mean of the predictions of the decision trees as one prediction. It is worth mentioning that different parts of the data are used to build up different decision trees by choosing n samples from the training set with replacement (i.e., random bootstrap samples of size n). This makes the RF regressor less sensitive to overfitting, in contrast to decision trees. Note that overfitting is an issue where the model learns the unnecessary characteristics of the training data so that it may not perform well on unseen test instances. 

XGB [52] is one of the leading supervised ML algorithms, widely used for building ML models implemented for both regression and classification tasks. The gradient boosting tree model identifies the flaws of the weak learners and boosts the gradient descent with each weak learner in the loss function. The loss function is calculated from the difference between the predicted value and the true value. Predictors in the ensemble correct the mistakes of the previous node until the leaf is reached. XGB model is selected for its scalability, performance, and computational speed as well as its ability to handle noise and variance. 

Typically, in machine learning, the percentage of the data division for training and test dataset is 80 and 20%, respectively. Accordingly, we used 30000 randomly chosen instances for training and the remaining 7500 instances for the test. To tune and assess the predictors, we reported their performance in terms of well-known regression metrics, such as explained variance regression score (EVS), root mean squared error (R^2^ Score), mean absolute error (MAE), and root of mean squared error (RMSE). Next, experimental optical parameters were given as input to each trained ML algorithm and their accuracy predictions were tested.

## 3. Results

Figure 4 presents the density distribution of the LF, ccx, and ccy in the test and training datasets. It can be seen that the dataset contains WLEDs with a wide range of brightness levels covering a wide spectrum of CIE color coordinates. After fitting the model by each predictor, their prediction performance as well as the plot of the relationship between the real and predicted junction temperatures is presented in Figure 5. Note that there are two types of data: data provided by manufacturers and data obtained through experiments. Figure 5 depicts the relationship between the prediction and real temperatures based on the information provided by the manufacturers on the left, whereas plots on the right depict the relationship based on data provided by the experiments. It is worth remembering that the former reflects the manufacturers’ desired/estimated temperatures, whilst the latter shows the actual temperatures that we measure during our tests. An agglomeration of data points can be noticed in the experimental data in the 80 °C region, given the fact that devices under test reach this temperature quickly. Referring to Figure 3, this is due to the abrupt spike in the temperature from room temperature in the test devices. In this period, the constant exposure time of the spectrophotometer results in the acquisition of fewer data points in comparison to the steady-state condition.

According to the results depicted in Figure 5, it can be inferred that the performance of the KNN predictor (with k = 2) is similar on both the manufactures’ data and experimental data in terms of the EVS and R^2^ score. Note that the near-to-unity R^2^ score means that the predictor is more successful in its predictions, while lower values of the MAE and RMSE are more desired in terms of the prediction accuracy. The model based on the manufacturers’ data has a slightly greater RMSE and MAE, possibly due to the size scale of the data. With the EVS and R^2^ scores near 1, and a lower RMSE score, the KNN model outperforms the RNN model. Especially, the EVS and R^2^ scores of the RNN are significantly lower on the experimental data. Although both models are similar in terms of training, the difference between the two models is in how the training dataset is used during prediction. The RNN selects within a given normalized radius rather than identifying the k-neighbors. The skewness in the density distributions appears to have resulted in inaccurate RNN predictions.

By looking at the plots and scores regarding the RF, it can be seen that the EVS and R^2^ scores of the RF are slightly lower than the KNN and RNN on the experimental data, while it is the other way around on the manufacturer’s data. Moreover, it performs better in terms of the RMSE and MAE on the manufacturer’s data but worse on the experimental data compared to the KNN and RNN. Finally, it can be seen that the performance of the XGB is much better than all the other predictors in terms of the RMSE and R^2^ score. The XGB models could achieve successful learning of the complex relationship between the input features and target temperatures. For instance, high-power and low-power chips (different LF) can present exactly the same ccx and ccy values and seemingly the XGB is more accurate in these instances.

Despite the fact that the WLEDs used in the experiments were from various chip manufacturers, the machine learning methods demonstrated high reliability in relating the simplified SPD-related parameters to the junction temperature. Even with the same bandgap characteristic in the chip, the difference in the phosphor characteristics can be expected in different packages. For instance, the used Yttrium Aluminum Garnet (YAG) phosphor from different manufacturers can be doped with different Ce^3+^ concentrations. The luminescence quenching temperature can vary depending on the doping or preparation characteristics of the used phosphor [53]. Even with this in mind, we can see a good predicting capability of ML algorithms to relate light to temperature characteristics in solid-state lighting packages. This method can be employed as a reliable and fast tool for junction temperature estimation by manufacturers, particularly in wafer-level probing or for covered/coated chips. Furthermore, simple imaging tools capable of recording the color characteristics and brightness levels of light sources can be equipped with these models to perform real-time temperature measurements for installed operational devices where conventional junction temperature measurement methods are not possible.

## 4. Conclusions

In this study, a new method for predicting the junction temperature of WLEDs based on optical parameters is proposed. A consolidated database of the color and brightness characteristics of WLEDs from two major manufacturers is amassed to train four different ML-based models. The models were tested experimentally with WLED packages from various manufacturers in order to demonstrate their capability. The XGB model has shown a better performance with an MAE of 0.525% with a close-to-perfect correlation capability. The basic LED junction temperature measurements are essential to each unit in the fabrication or application process. Conventional inspection methods require large measurement equipment (and in manufacturing, a high-precision alignment stage). This becomes an issue in the case of micro-LEDs or coated LEDs. The method herein can utilize a single camera to inspect the brightness and color characteristics of the unit under test and simultaneously provide a real-time prediction of its junction temperature. The results herein are restricted to the nominal forward current density of 35 A/cm^2^, but the procedure can be extended to other current densities.

## Figures and Tables

**Figure 1 micromachines-13-01245-f001:**
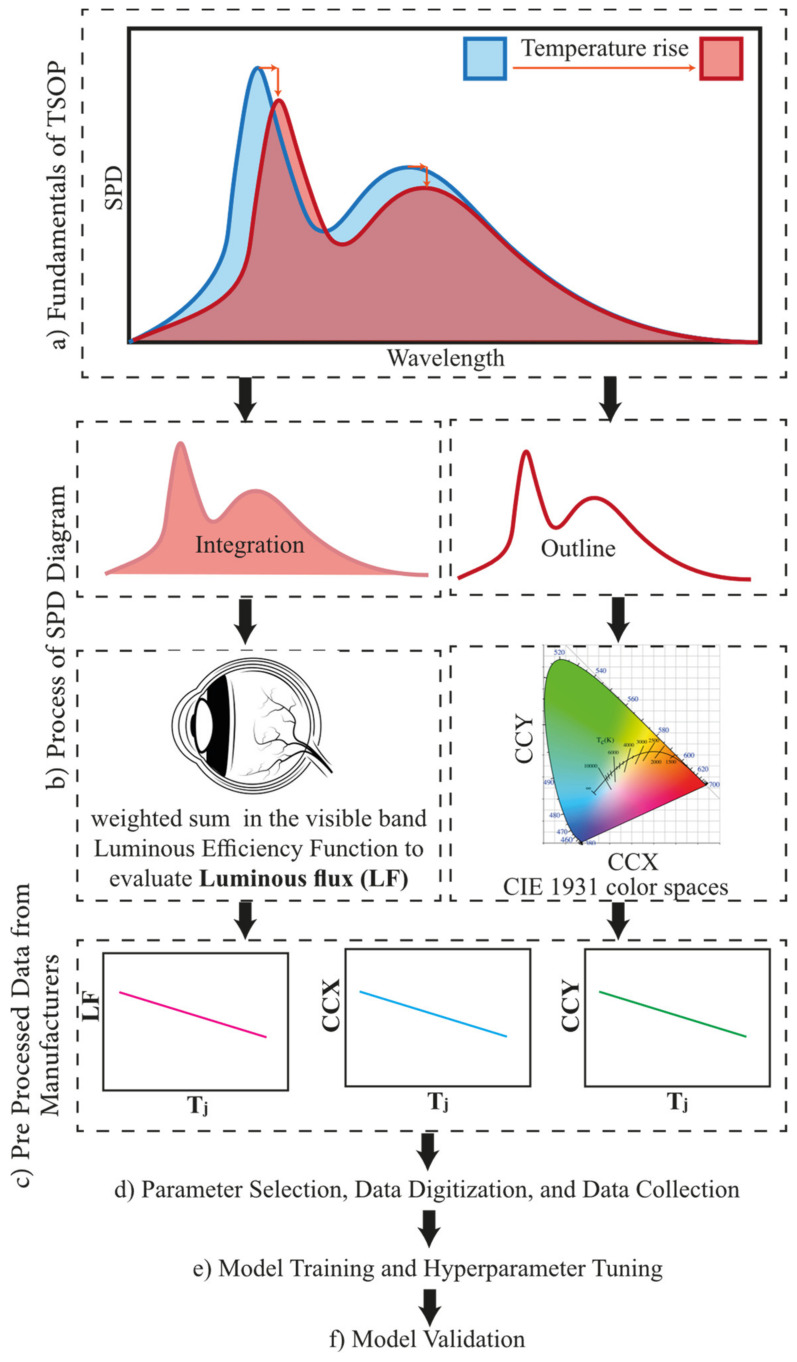
Road map illustration of the current study which provides (**a**) basic behavior of the spectral power distribution (SPD) of a WLED by the temperature rise of the junction, (**b**) process of the SPD diagram for luminous flux (LF), and correlated color temperature (CCT) determination. Based on this processed information, manufacturers provide (**c**) postprocessed data for the determination of the temperature sensitivity of the chip. (**d**) Provided data from manufacturers can be digitized and collected for (**e**) ML model training and tuning. (**f**) Model validation for the accuracy of the universal algorithm.

**Figure 2 micromachines-13-01245-f002:**
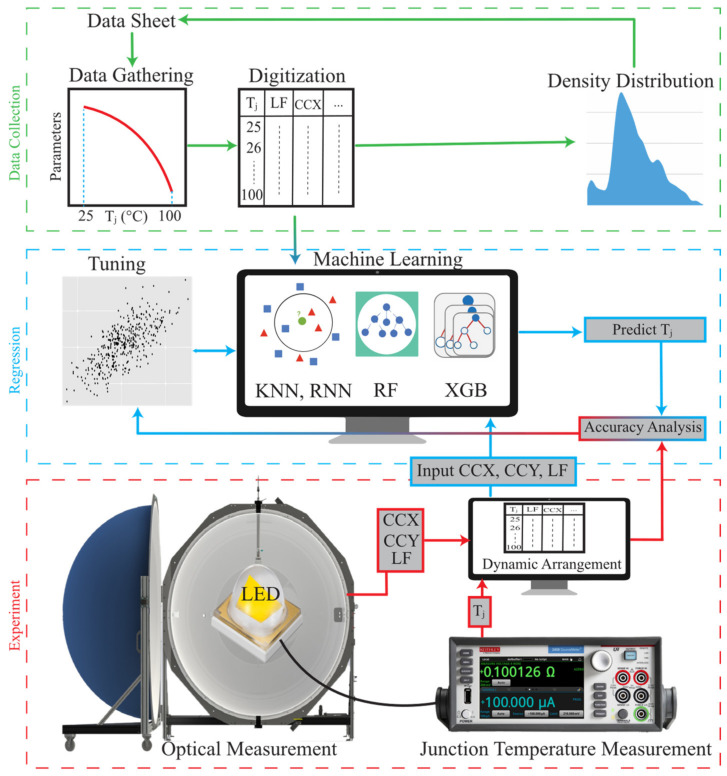
Overview of data collection, regression, and experiments and their interconnection.

**Figure 3 micromachines-13-01245-f003:**
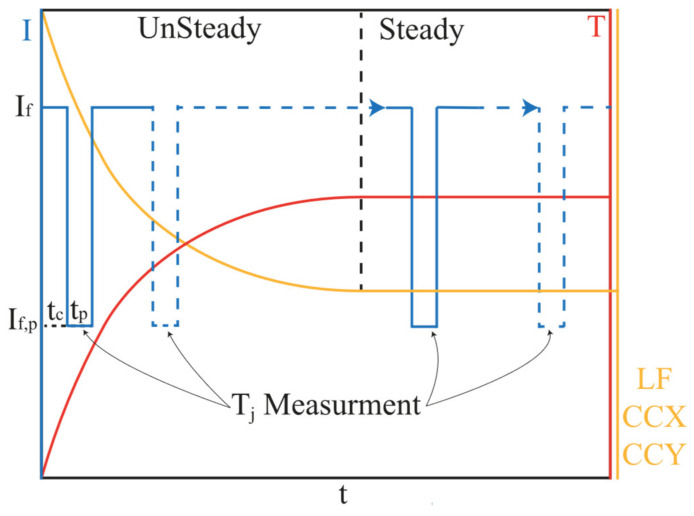
Illustration of the transient test phase (red curve shows the trend of temperature rise of the junction, and the yellow curve assimilates the trend of optical parameters).

**Figure 4 micromachines-13-01245-f004:**
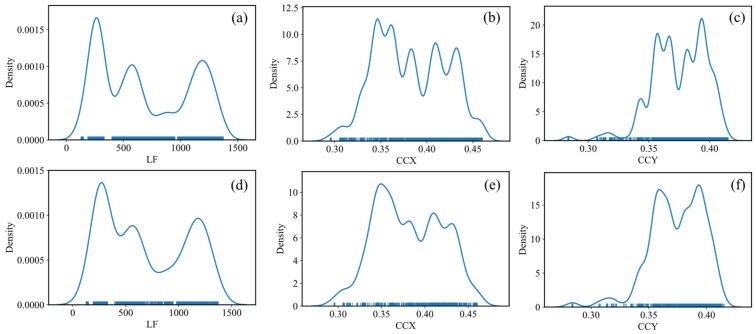
Density distribution of (**a**) LF, (**b**) CCX, and (**c**) CCY in training dataset. Density distribution of (**d**) LF, (**e**) CCX, and (**f**) CCY in test dataset.

**Figure 5 micromachines-13-01245-f005:**
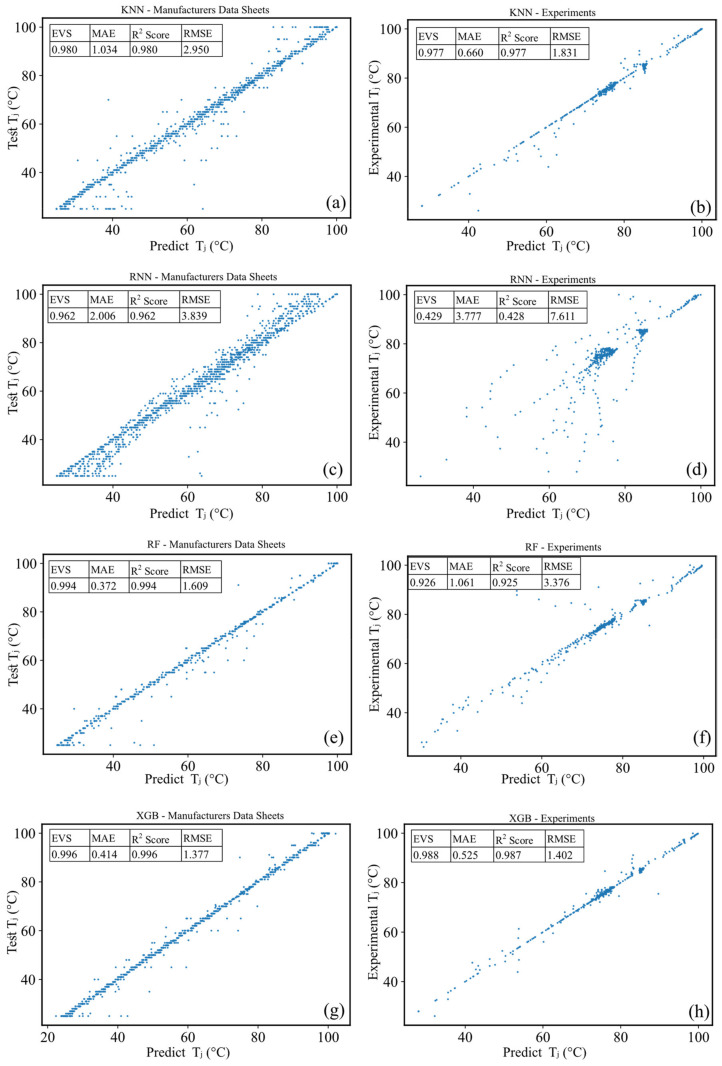
(**a**) KNN, (**c**) RNN, (**e**) Rf, and (**g**) XGB predictions based on package data sheets. (**b**) KNN, (**d**) RNN, (**f**) RF, and (**h**) XGB predictions based on experimental inputs. EVS, MAE, R^2^ score, and RMSE for each case are provided in a subtable.

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
