# Peer review of "Machine Learning to Predict Junction Temperature Based on Optical Characteristics in Solid-State Lighting Devices: A Test on WLEDs"

_micromachines, 2022, doi:10.3390/mi13081245_

Round 1
Reviewer 1 Report
The manuscript entitled “Machine Learning to Predict Junction Temperature Based on Optical Characteristics in Solid State Lighting Devices: A Test on WLEDs” was interesting and well written by the authors. The authors clearly described how to predict the junction temperature of WLEDs based on luminescence parameters and the advantages of their proposed method over the conventional junction temperature measurements. Also, the authors tested the WLED packages from various manufacturers. Therefore, in my view, the manuscript is suitable for publication in its present form.
Author Response
Dear Editor,
We appreciate the opportunity to propose the revised draft of the “Machine Learning to Predict Junction Temperature Based on Optical Characteristics in Solid State Lighting Devices: A Test on WLEDs” for publication in the Journal of Micromachines. We are, again, thankful for the valuable time and effort provided from you and the comments of the reviewer on further improvement of the manuscript. In our revised document, we have tried to incorporate as much as we can to cover all the suggestions provided by the reviewers and editorial team. To ease of following any added part to the manuscript is highlighted with the review capability of Microsoft Word. Please find below, in tables, for a point-by-point response to the reviewer’s comments and concerns.
Best Regards
Mohammad Azarifar
|
Review Comments |
Author’s Response |
| The manuscript is suitable for publication in its present form. | Thank you. |
Reviewer 2 Report
Dear autor,
you investigate the prediction of LEDs junction temperature by the use of ML on optical characteristics. A very important intensity that could definitely find application in the industrial environment as the detection of temperature plays an important role for LED qualility test as well as reliability test. In any case, the process should be further pursued and optimized in order to make the application usable for the different types of LEDs.
Notes:
line 212, 297 - use only one convention "Fig." or "Figure", but do not mix it.
line 213, 242-261, 300-301, 320, 332 348 - use passive voice instand of "we"
line 296 - "R"esults
Regards, the Reviewer
Author Response
Dear Editor,
We appreciate the opportunity to propose the revised draft of the “Machine Learning to Predict Junction Temperature Based on Optical Characteristics in Solid State Lighting Devices: A Test on WLEDs” for publication in the Journal of Micromachines. We are, again, thankful for the valuable time and effort provided from you and the comments of the reviewer on further improvement of the manuscript. In our revised document, we have tried to incorporate as much as we can to cover all the suggestions provided by the reviewers and editorial team. To ease of following any added part to the manuscript is highlighted with the review capability of Microsoft Word. Please find below, in tables, for a point-by-point response to the reviewer’s comments and concerns.
Best Regards
Mohammad Azarifar
|
Review Comments |
Author’s Response |
|
line 212, 297 - use only one convention "Fig." or "Figure", but do not mix it. |
Thank you for suggestion. We adapted the Fig. in the manuscript.
|
|
line 213, 242-261, 300-301, 320, 332 348 - use passive voice instand of "we"
|
Thank you for suggestion, the passive sentences are considered in the manuscript in the method and result sections. |
| line 296 - "R"esults |
Thanks for noticing, this is probably happened in converting the word document in MDPI server and might happen again when we upload the new document, which we can fix it again in proof reading process. |
